# Current Therapeutic Options in the Treatment of Rheumatoid Arthritis

**DOI:** 10.3390/jcm8070938

**Published:** 2019-06-28

**Authors:** Birgit M. Köhler, Janine Günther, Dorothee Kaudewitz, Hanns-Martin Lorenz

**Affiliations:** Internal Medicine 5, Division of Rheumatology, University Hospital Heidelberg, 69120 Heidelberg, Germany

**Keywords:** Rheumatoid Arthritis, therapy, DMARD, MTX, Tumor Necrosis Factor-Alpha Inhibitors

## Abstract

Rheumatoid arthritis (RA) is a systemic autoimmune disease characterized by chronic inflammation of the joints. Untreated RA leads to a destruction of joints through the erosion of cartilage and bone. The loss of physical function is the consequence. Early treatment is important to control disease activity and to prevent joint destruction. Nowadays, different classes of drugs with different modes of action are available to control the inflammation and to achieve remission. In this review, we want to discuss differences and similarities of these different drugs.

## 1. Introduction

Rheumatoid Arthritis (RA) is an autoimmune disease characterized by chronic inflammation of the synovial membrane. Untreated RA can lead to progressive joint destruction, resulting in disability, poor quality of life, and increased mortality. About 1% of the population is affected, and the disease onset generally occurs between 30 years and 50 years of age, with a higher incidence in women.

The therapy is complex and includes different classes of drugs with different routes of application but also nonpharmacologic interventions. The most important are patient education followed by exercise and physical and occupational therapy. Because of an increased risk of coronary atherosclerosis, efforts should be made to reduce risk factors such as smoking, hyperlipidemia, hypertension, and obesity.

To relieve pain and swelling fast and to gain control of the inflammation, glucocorticoids (GC) are used widely in acute disease flares either orally or as intraarticular injections. Oral GC is for short-term use (up to 3–4 month) only and should be tapered to prevent side effects as soon as possible [1]. To control inflammation in the long run, Disease Modifying Anti-Rheumatic Drugs (DMARD) to spare GC are needed. Nowadays, there are a bunch of opportunities that can be challenges or chances.

The treatment of patients with RA aims to relieve pain and to control inflammation, and the final goal is to achieve remission or at least low disease activity for all patients. In this context, the European League Against Rheumatism (EULAR) has composed 10 international recommendations on how to treat patients [2]. An algorithm based on the EULAR recommendations is shown in Figure 1. In 2010, an international committee developed the treat-to-target (T2T) initiative. The centerpiece of this initiative is the shared decision-making and regular patient revaluation that targets remission or at least low disease activity (LDA) [3].

By now, there is evidence from different studies that the T2T principle is superior, and it forms part of the treatment guidelines of the European League Against Rheumatism and the American College of Rheumatology. The core principles of T2T are shown in Table 1. Even more important is that an early start of therapy is required in order to achieve optimal outcomes.

## 2. Treatment Guidelines: Conventional Synthetic DMARD (csDMARD)

As soon as the diagnosis of rheumatoid arthritis is made, a treatment with a csDMARD should be started [4].

A controlled comparison between csDMARDs for the first-line therapy does not exist; however, within this group, methotrexate should be the first choice because, for this drug, most clinical experience exists in monotherapy and as a combination partner with other DMARDs [5]. Methotrexate is usually started at a dose of 15 mg/week and can be stepwise increased up to 25 mg/week. The combination with glucocorticoids is recommended [3]. Due to the decreased bioavailability, a subcutaneous way of administration is recommended [5].

The induction of remission with a combination of conventional synthetic DMARDs at this stage is not superior to methotrexate monotherapy; however, these combinations are associated with more adverse events and a higher rate of drug discontinuation [4,6]. Patients with a higher baseline disease activity and Rheumatoid factor (RF)-positive patients have an increased risk of methotrexate (MTX) failure due to inefficacy [6].

If MTX cannot be used, e.g., due to intolerance or contraindications, leflunomide (20 mg/week) or sulfasalazine (2 g/day) should be started. In a placebo-controlled randomized controlled trial (RCT), both substances showed a similar efficacy [7].

If by week 12 after the start of MTX therapy no adequate response is achieved or no remission is reached with optimum doses after 24 weeks, the therapy should be adjusted [5]. To find the best individual treatment strategy, patients should be categorized using prognostic markers. Poor prognostic markers such as the presence of autoantibodies, early joint damage, and high disease activity are associated with rapid disease progression; therefore, a biologic DMARD or a targeted synthetic DMARD should be added at this stage [4]. In the absence of poor prognostic markers and with moderate disease activity, a second csDMARD should be added to the therapy [5].

### 2.1. MTX

Unlike targeted DMARDs, conventional synthetic DMARDs came into clinical practice based on empiric observations and their mechanisms of action are still incompletely understood [4].

The mode of action of high-dose methotrexate via the depletion of thymidine and purine residues and cell-cycle arrest at S1 are well-known [8]. However, this mechanism does not seem to play a major role in the clinical effect of low-dose MTX, as folate co-therapy does not result in a loss of clinical benefit. Methotrexate has pleiotropic therapeutic effects on various immune cells and mediators, resulting in an overall dampening of the inflammatory response. The main mode of action of low-dose MTX in rheumatoid arthritis is thought to be via the potentiation of adenosine signaling. Adenosine acts as a paracrine signaling agent via four distinct purinergic G-protein-coupled receptors, which in rheumatoid arthritis are overexpressed. In addition to downregulating the production of tumor necrosis factor (TNF) and NF-kB, adenosine might be one of the main mediators of the downregulation of the activation and proliferation of T-lymphocytes, creating an immunotolerant environment [8].

Side effects of MTX are usually dependent on dose, mode of application, and duration of methotrexate therapy. Unlike high-dose MTX, the side effects of low-dose MTX are rarely life-threatening and can often be relieved by substituting folate. Common side effects of low-dose MTX are hematologic abnormalities (thrombocytopenia and leucopenia), stomatitis, gastrointestinal problems (e.g., anorexia, loose stools, nausea, or stomach upset), elevation of liver enzymes, or central nervous system symptoms like fatigue or headache [9]. The substitution of folate significantly lowers the risk for side effects like hepatotoxicity (relative risk reduction of 77%) and decreases the number of serious adverse events (by 61%) [5,10]. If side effects occur, substituting folate acid regularly and gradually increasing the dose up to 5 mg daily can help to control the symptoms. Overall, fewer than 5% of patients have to stop using methotrexate because of adverse events [4].

MTX therapy can, in many cases, be safely continued perioperatively; however, a potentially decreased kidney function in this setting should be taken into account. In addition, to decrease the risk of pneumonia, the treatment should be paused if pulmonary comorbidities exist. At high doses (25 mg/week), a temporary dose reduction should be considered [11].

MTX exposure during pregnancy can induce multiple congenital deformities. Therefore, MTX therapy during pregnancy is not recommended. MTX should be withdrawn prophylactically 3 months before conception. Daily folate supplementation should be continued antenatal and throughout pregnancy. At the present stage of knowledge, it is not clear whether MTX transiently influences male fertility and sperm DNA integrity [12].

### 2.2. Leflunomide

The primary mechanism of action of leflunomide (LEF) is the reversible inhibition of the mitochondrial enzyme dihydroorotate dehydrogenase (DHODH), the rate limiting step in the de novo synthesis of pyrimidines. Activated lymphocytes expand their pyrimidine pool by approximately eightfold during proliferation. Therefore, the inhibition of DHODH prevents activated lymphocytes from moving from G1 to the S phase, hence triggering apoptosis [13]. The leflunomide effect seems to be rather lymphocyte-specific on other cells, can reuptake pyrimidines, and can thereby overcome the DHODH blockade.

The rate of discontinuation due to side effects is similar with methotrexate [14]. Potential side effects include diarrhea and nausea as well as the elevation of liver enzymes. The changes in liver function are generally reversible with dose reduction or a discontinuation of the drug, but in rare cases, hepatotoxicity can be severe. However, transaminase elevation mainly occurs if other comorbidities contributing to hepatotoxicity, e.g., concomitant non-steroidal anti-inflammatory drugs (NSAID) or MTX therapy, previous or concurrent alcohol abuse, or viral or autoimmune hepatitis, are present [15]. A small percentage of patients with RA develop hypertension when taking LEF; therefore, blood pressure monitoring is recommended during treatment [14].

The active metabolite of leflunomide is detectable in plasma until 2 years after the discontinuation of the drug; therefore, a discontinuation of leflunomide perioperatively is generally not recommended. Only at a high risk of infection or if a greater intervention is planned, a wash out with cholestyramine should be initiated as recommended [11].

Leflunomide is contraindicated during pregnancy. Safe contraception during therapy in both women and men is recommended. Before conception, leflunomide must be withdrawn and a washout should be carried out until the drug is undetectable in the blood [12].

### 2.3. Sulfasalazine

Sulfasalazine SSZ was specifically designed in 1938 for the treatment of rheumatoid arthritis. The idea behind the drug was to combine an antibacterial and an anti-inflammatory agent [16]. Sulfasalazine is effective in the treatment of rheumatoid arthritis; however, the mode of action is incompletely understood. The main pharmacological effects of SSZ include effects on the gut bacterial flora, on inflammatory cell functions, and on immunological processes [16]. Several plausible mechanisms of action have been observed in vitro, such as the inhibition of NF-kB and osteoclast formation via modulatory effects on the receptor activators of NF-kB (RANK), osteoprotegerin (OPG), and RANK-ligand. In addition, SASP can inhibit tumor necrosis factor (TNF)-alpha expression and may reduce the secretion of inflammatory cytokines such as interleukin (IL)-8 as well as may suppress B-cell function [17]. An additional mechanism that has been suggested is the increased production of adenosine at sites of inflammation similar to the mode of action of methotrexate.

Adverse reactions, including idiosyncratic (e.g., hypersensitivity-/immune-related) and dose-related effects, are common with sulfasalazine, especially gastrointestinal, central nervous system, cutaneous, and hematologic adverse effects. The withdrawal rate for adverse events is about 25%, two thirds of which are due to gastrointestinal and central nervous system toxicity. If dose-related side effects occur, treatment can be paused for a week and after a resolution of the symptoms, the treatment can be restarted at a lower dose [18]. However, if idiosyncratic effects like skin reactions, hepatitis, pneumonitis, or hematologic side effects like agranulocytosis and hemolytic anemia occur, immediate discontinuation of the drug is necessary; patients with this type of adverse effect should not be rechallenged with the drug [18].

As sulfasalazine only has a short half-life of about 4–5 h and only a minimal immunosuppressive effect, it can usually be continued perioperatively. If there is a risk of interaction or a potential additive hepatotoxic effect with medication used preoperatively, sulfasalazine can be paused on the day of the operation [11].

If treatment of rheumatoid arthritis is required during pregnancy, SSZ is an acceptable therapeutic option, as the continuation of SSZ during pregnancy is very unlikely to cause fetal harm [12,19]. To increase safety, a concomitant folate supplementation before and throughout pregnancy is advised and the dose of SSZ should not exceed 2 g per day to prevent neutropenia in the newborns. SSZ can cause reduced male fertility; however, spermatogenesis recovers at about 2–3 months after a withdrawal of the drug [12].

## 3. Biologic DMARDs (bDMARD)

According the EULAR guidelines, a bDMARD should be considered if remission or LDA is not achieved with the first DMARD strategy, if poor prognostic factors (i.e., high acute phase reactant levels, high swollen joint counts, or the presence of early erosions) exist, or if the patient responds inadequately to MTX and/or other csDMARD strategies.

Before starting a therapy with bDMARD, active or latent infections with hepatitis or tuberculosis must be ruled out. Patients, if possible, should be brought up to date with all immunizations before initiating therapy. Blood cell counts and liver and kidney function also need to be evaluated prior to treatment. 

### 3.1. Tumor Necrosis Factor-Alpha Inhibitors (TNFi)

Until now, five TNFi were available. Although all anti-TNF drugs bind TNF-α, there are differences in their molecular structures, their administration regimens, and their modes of action.

The first preclinical studies using antibodies against TNF- α were performed in animal models of sepsis in 1985. About 6 years later, Keffer et al. provided the first evidence that TNF plays a role in developing arthritis [20]. In 1994, *The Lancet* published the first RCT showing that the blockade of a specific cytokine can be an effective treatment in patients with rheumatoid arthritis [21]. Since then, antibodies against TNF-alpha have gotten an important cornerstone in the treatment of RA and changed the lives of our patients.

Nowadays, we could not think of a world without antibodies against TNF-α or other cytokines. For our patients, this changed the game.

The first TNF-α-inhibitor, approved in 2000 for the therapy of RA, was the chimeric murine/human IgG1 monoclonal antibody Infliximab (IFX) that binds to both soluble and membrane-bound TNF-α [21]. The administration is intravenous every 8 weeks at a dosage between 3–5 mg/kg. Another TNF-inhibitor also approved in 2000 is Etanercept (ETN), a recombinant fusion protein compound of the soluble TNF-alpha receptor linked to the Fc portion of human IgG. ETN binds to the TNF receptor, preventing TNF-mediated cellular responses. ETN is administered subcutaneously at a dose of 25 mg twice a week or 50 mg weekly [22,23]. The third inhibitor approved in 2003 by the European Medicines Agency (EMA) is Adalimumab (ADA) [24]. It is a recombinant human IgG1 monoclonal antibody that binds to soluble and membrane-bound TNF-α with a high affinity. It is administered by subcutaneous injection once every 2 weeks. Golimumab (GOL) is a human IgG1 monoclonal antibody neutralizing both soluble and membrane-bound TNF-α. GOL was approved in 2010 and is administered as a subcutaneous injection at an initial dose of 50 mg every 4 weeks that can be increased to 100 mg if there is no response after 4 doses (in patients with a body weight > 100 kg) [25]. Certolizumab Pegol (CZP) was approved by the EMA in 2007. CZP is a recombinant humanized Fab’ fragment of a TNF antibody coupled to polyethylene glycol (PEG) that prolongs its plasma half-life to approximately 2 weeks. It has an initial loading dose of 400 mg every 2 weeks for 6 weeks, followed by 200 mg every 2 weeks [26,27].

Only certolizumab, adalimumab, and etanercept are approved as monotherapy as well as in combination with methotrexate [27,28,29,30,31].

Adverse effects include skin reactions characterized by itching, pain, and redness at the site of medication injection. Such injection site reactions characteristically occur during the first weeks of treatment. For intravenously administered agents, acute infusion reactions can occur with hypotension, bronchospasm, wheezing, and/or urticaria. Acute reactions may, in some cases, represent immunoglobulin-E-mediated type I reactions. The majority of acute infusion reactions that occur are anaphylactoid reactions and not immunoglobulin E mediated. These reactions can be managed by just reducing the rate of infusion [32,33].

However, delayed infusion reactions can also occur, and they are associated with skin rash, diffuse joint pains, myalgia, and fatigue and sometimes accompanied by fever. Delayed reactions may represent mild type III (immune complex-mediated) reactions [34]. It has been shown that the formation of Anti-monoclonal antibodies may lead to a greater risk of infusion reactions and also may limit the long-term efficacy of the drug. It has also been shown that the use of concomitant immunomodulators prior to starting a TNF-α-inhibitor is effective in reducing antibody production and therefore decreasing immunogenicity [32,34].

As TNF-alpha is a key player of the immune system during infections, this treatment has been associated with an increased risk of serious infections. These include bacterial infections (particularly pneumonia), herpes zoster, tuberculosis, and opportunistic infections [35]. As mentioned above, screening for latent tuberculosis infections should be performed before the initiation of TNF-alpha inhibitor therapy. If there is an indication of latent tuberculosis, a treatment for latent tuberculosis should be initiated before starting therapy with a TNF-alpha inhibitor [36]. Another side effect occurring in a few cases is neutropenia. Other cytopenias are uncommon [37]. The association between TNF-alpha inhibitors and demyelinating diseases remains uncertain. However, anti-TNF-alpha agents should generally be avoided in patients with established diseases that are associated with demyelination and should be discontinued directly in any patient with suspected demyelination [38]. Treating patients with NYHA III or IV cardiac failure is not recommended [2,39]. However, data regarding cardiac failure in patients treated with anti-TNF-alpha agents are inconclusive [40]. While Kwon et al. reported 38 patients who developed new-onset heart failure and 9 patients who experienced heart failure exacerbation after therapy [39], others report a reduced risk of cardiovascular events in patients treated with TNF-alpha inhibitors [40].

#### Pregnancy

Observational studies and case reports of women exposed to TNF inhibitors during pregnancy suggest that their pregnancy outcomes, including rates of preterm birth, spontaneous miscarriage, and congenital anomalies, are similar to those in women with RA who have not received a TNFi [41].

Complete immunoglobulin G (IgG) antibodies—maternal as well as therapeutic—cross placenta via active transport facilitated by the neonatal fragment crystallizable (Fc) receptor on the placenta. IgG1 is the most effectively transported of the four subclasses of IgG (G1–G4) [42]. This is relevant to adalimumab, golimumab, and infliximab which are complete IgG1 anti-TNF. Etanercept is comprised of the Fc domain of human IgG1 fused with the extracellular ligand binding domain of human tumor necrosis factor receptor-2. Transplacental transport via the neonatal Fc receptor would theoretically be plausible. In contrast to the complete IgG1 anti-TNF antibodies certolizumab pegol differs structurally as it is a humanized PEG (polyethylene glycol)-ylated antibody Fab’ fragment lacking the IgG1 Fc portion. Without the Fc portion it should, in theory, not be transported actively across the placenta [43]. So passive diffusion would be the only option for any detectable concentrations in exposed infants. This was shown in two trials. Based on these trials the EMA has approved a label change for CZP, making it the first anti-TNF for potential use in women with chronic rheumatic disease during both pregnancy and breastfeeding [44].

TNFi may have different structures, morphology, pharmacokinetic properties, and activity, but their clinical efficiency is comparable. If drug survival and safety are similar or different is still a matter of debate.

### 3.2. Interleukin 1 Inhibitor

Anakinra was first approved in the US in 2001 and in Europe in March 2002. It is a recombinant human IL-1 receptor antagonist with a short half-life (4–6 h), administered subcutaneously at a dose of 100 mg once a day. It was first developed for use in RA and showed some effects in early trials. A big systematic review of the literature in 2009 showed only a moderate effectiveness [45,46]. Anakinra plays not a very great role in RA therapy; today, it is much more effective in the therapy of auto-inflammatory diseases, gout, or polyserositis [45,46,47,48,49,50].

Adverse effects include injection site reactions characterized by itching, pain, and redness at the site of medication injection and lasting days to weeks. Between one and ten percent of people have severe infections, decreased white blood cells, or decreased platelets [51].

#### Pregnancy

There have been no adequate studies for the safety of Anakinra in Pregnancy, although several case reports of women with adult-onset Still’s disease treated through pregnancy resulted in healthy neonates [52].

## 4. Interleukin 6 and Interleukin 6 Receptor Inhibitors

Interleukin-6 (IL-6) was identified in 1986 as a key cytokine in the pathogenesis of RA with proinflammatory activity. It is able to enhance the production of acute-phase proteins involved in the systemic inflammation process. Tocilizumab (TCZ) is the first humanized recombinant IgG1monoclonal antibody that binds to both the soluble and membrane-bound IL-6 receptor, blocking its action and leading to the decrease of the inflammatory response cascade. Its half-life (10–13 days) allows its administration intravenously (8 mg/kg) every 4 weeks. A subcutaneous formulation of TCZ (162 mg/week) has been approved, with an efficacy and safety profile comparable to intravenous administration [52,53]. A second agent, Sarilumab is a human immunoglobulin G1 anti-interleukin-6 (IL-6) receptor monoclonal antibody that blocks IL-6 from binding to membrane-bound and soluble IL-6 receptor alpha [54].

Events of gastrointestinal (GI) perforations have been reported in Phase III clinical trials, primarily as complications of diverticulitis, including generalized purulent peritonitis, lower GI perforation, fistula, and abscess. Tocilizumab or Sarilumab should be used with caution in patients who may be at increased risk for GI perforation. Patients presenting with new-onset abdominal symptoms should be evaluated promptly for the early identification of GI perforation. It is important to keep in mind that, during IL-6 blockade, C-reactive protein or other acute phase reactants will increase slowly and less pronounced [55,56].

Other side effects are neutropenia or thrombocytopenia as well as hyperlipidemia. Also, serious infections and liver enzyme elevations were observed and can require dose adjustments or drug discontinuation [56].

### 4.1. Pregnancy

Based upon animal data, tocilizumab may cause fetal harm, but there are no adequate studies of its effects on human pregnancy to allow an assessment of its risk. There are a few case series available. No teratogenic effects were observed. In a case series of 50 pregnancies exposed to tocilizumab with known outcomes, 36 resulted in live births, while there were five low-birthweight infants born and one case of neonatal asphyxia [57,58].

### 4.2. CD80/86-CD 28 Inhibitor

Abatacept is a fusion protein constituted by an immunoglobulin fused to the extracellular domain of cytotoxic T-lymphocyte antigen 4 (CTLA-4). This is a molecule that binds with a high affinity to the CD80/86 ligand on antigen-presenting cells. Therefore, abatacept is able to block the interaction between the antigen-presenting cell’s CD80/86 ligand and the CD28 ligand on the T cell [59]. This results in decreased T cell proliferation and cytokine production. Abatacept is administered intravenously once every 4 weeks or subcutaneously once a week [60,61].

Adverse effects including cases of hypersensitivity and anaphylaxis or anaphylactoid reactions have been reported with iv administration. As in other bDMARDs, serious infections (including tuberculosis and sepsis) have been reported, particularly in patients receiving concomitant immunosuppressive therapy [59].

The use of CTLA-4 Inhibitors due to its T cell inhibition affect defenses against malignancies. As compared to the general population, an increased risk of lymphoma has been noted in clinical trials; however, rheumatoid arthritis has been previously associated with an increased rate of lymphoma. Abatacept itself seems not to further increase this risk [62].

#### Pregnancy

Abatacept is not teratogenic in animals, but there are no adequate studies of its effects on human pregnancy to allow an assessment of its risk.

## 5. Anti-CD 20 Antibody

Rituximab (RTX) was initially developed for the treatment of hematologic malignancies. Since 2006, RTX is approved for the therapy of RA refractory to a combination therapy of anti-TNF-alpha, an MTX [60]. RTX is a chimeric murine-human monoclonal antibody that binds to the CD20 membrane receptor, leading to the depletion of circulating B cells. It is also able to inhibit the activation of T cells that produce proinflammatory cytokines. A cycle of RTX consists in 1000 mg by intravenous infusion, followed by a second 1000-mg intravenous infusion two weeks later. This is then repeated every 6 months. Different studies showed that particularly patients who are autoantibody-positive benefit from RTX [63,64].

According to results of different studies, patients benefit more from switching to RTX when an initial TNFi was discontinued due to inefficacy.

One of the common side effects is an infusion reaction especially during the first infusion, occurring in up to 30 to 45 percent of patients. Symptoms include headache, fever, skin rash, dyspnea, hypotension, nausea, rhinitis, pruritus, and mild angioedema [65].

Repeated courses of RTX are associated with an increasing risk of hypogammaglobinemia. A meta-analysis in 2009 found no increase in serious infections associated with the use of rituximab with or without MTX compared with MTX plus a placebo, but other studies have found that repeated courses may be associated with a higher rate of serious infections. The risk of serious infections seems to increase with age [65].

### Pregnancy

There is only limited information about the use of RTX during pregnancy. Rituximab has been detected in high concentrations in umbilical cord blood. There are case reports available, but in most cases, the mother took RTX due to hematologic malignancies. Congenital anomalies were not reported. As for all babies indirectly exposed to immunosuppressant biologics during pregnancy, immunization with live vaccines should be postponed until the age of 6 months [66].

## 6. Targeted Synthetic DMARDS: JAK-Inhibitors

Recently, with the Janus–Kinase (JAK) Inhibitors, a new group of drugs for the treatment of rheumatoid arthritis was introduced. These targeted, synthetic DMARDS (tsDMARDS) are effective in inflammatory diseases by intracellularly blocking tyrosine kinase [67].

The JAK are cytoplasmic protein tyrosine kinases that are critical for signal transduction to the nucleus from the common gamma chain of the plasma membrane receptors for interleukin (IL)-2, -4, -7, -9, -15, and -21. JAK are receptor-associated intracellular proteins, with a tyrosine-kinase component. They are important downstream mediators of many pro-inflammatory cytokines, e.g., interferons or interleukin 6. Once a ligand binds to its receptor, the intracellular kinase is phosphorylated, which further leads to the phosphorylation and activation of the signal transducer and activator of transcription (STAT)-pathway. Accordingly, blocking these enzymes with a targeted molecule affects multiple inflammatory pathways [65,66]. How JAKs transmit signals from cytokines and thereby activates gene transcription is shown in Figure 2.

Pharmacological aspects: JAK-inhibitors are referred to as small molecules, meaning that these compounds carry a low molecular weight and bind to a macromolecular target, altering their activity and function [68,69].

To date, two JAK-inhibitors are approved for therapeutic use in rheumatoid arthritis: Tofacitinib and Baricitinib.

Tofacitinib was approved by the FDA (Food and Drug Administration) in the US in 2012. In 2017, the European Medicines Agency (EMA) approved the drug for use in the European Union. Tofacitinib is an inhibitor of JAK 1 and 3, with only little affinity to JAK 2 and Tyrosine-Kinase 2. It has been shown effective in moderate and severe rheumatoid arthritis in monotherapy or in combination with methotrexate. Several phase III clinical trials showed efficacy in DMARD naïve patients as well as in patients with insufficient responses to csDMARDs and even bDMARDS. Tofacitinib is taken orally twice daily with a dosage of 5 mg. The elimination is mostly hepatic; therefore, an adaption to impaired kidney function is not necessary up to a GFR > 30 mL/min. Consequently, the treatment of patients with end-stage renal failure with a GFR < 30 mL/min with Tofacitinib is possible with reduced dosage [70,71].

The second JAK-inhibitor currently approved is Baricitinib which is an inhibitor of JAK 1 and 2. It was approved by the FDA as well as the EMA in 2017. Baricitinib can be administered as monotherapy or in combination with methotrexate. Several phase III clinical trials showed efficacy in the treatment of DMARD naïve patients as well as in patients with insufficient responses to csDMARDS or even bDMARDS [72,73,74,75].

Baricitinib is taken orally once daily. There are two dosages available: 4 mg or 2 mg. The elimination for Baricitinib is mostly renal, which makes it necessary to reduce the dosage for patients with impaired kidney function (estimated GFR 30–60 mL/min) from 4 mg to 2 mg daily. For patients with an estimated GFR < 30 mL/min, the use of Baricitinib is not recommended [2].

According to the EULAR-Guidelines, both JAK-Inhibitors can be used in rheumatoid arthritis once a therapy with csDMARDs has been insufficient or had to be discontinued due to adverse events. Both Tofacitinib and Baricitinib have been shown to be more efficient than a placebo in the treatment of rheumatoid arthritisl; both could show non-inferiority compared with the TNF-inhibitor Adalimumab [2,71,73].

### Side Effects

Tofacitinib: Increased susceptibility to infections including especially herpes zoster are the most common side effects of Tofacitinib as well as headaches, hypertension, nausea, and diarrhoea. Elevated levels of low-density lipoprotein (LDL), high-density lipoprotein (HDL), total cholesterol, and liver enzymes have also been reported. In a post-marketing study, higher dosages of Tofacitinib showed an increased risk of deep vein thrombosis and pulmonary embolism. The recommended dosage of 5 mg twice daily should, therefore, not be exceeded [76,77].

Baricitinib: The most common side effect of Baricitinib was an increase of cholesterol levels. Elevated cholesterol levels return to normal once the medication is paused or be treated with statins effectively. This has been seen with other immunosuppressants as well. Other side effects include upper respiratory tract infections and nausea. Other infections, such as herpes zoster or pneumonia are also associated to Baricitinib. Patients treated with Baricitinib (0.3%) show significant neutropenia [78,79].

Screening pre-treatment: Active or latent infections with hepatitis or tuberculosis should be ruled out before initiating treatment with JAK-inhibitors. Blood cell counts and liver and kidney function also need to be evaluated prior to treatment.

Screening during treatment: neutrophil levels, lymphocyte levels, hemoglobin, and kidney and liver function should be screened; 8–12 weeks after the initiation of treatment, HDL, LDL, overall cholesterol, and triglycerides should also be screened.

Contraindications: JAK-inhibitors should not be prescribed for patients with neutropenia (<1/nL), active tuberculosis or severe infections, or severe liver impairment and during pregnancy. A combination with other bDMARDs or strong immunodepressants as azathioprine, cyclosporine, or tacrolimus should be avoided due to the elevated risk of infection and the lack of experience.

Use during Pregnancy: drugs from the JAK-inhibitor family have not been tested in pregnant women yet. As both drugs are small molecules, it seems likely that cross the placenta. Women taking JAK-inhibitors should be instructed to use safe contraceptive methods throughout the treatment and up to one week (Baricitinib) to four weeks (Tofacitinib) after a discontinuation of the drug.

Perioperative management: Although there are not enough long-term data for a conclusive recommendation on perioperative management, JAK-inhibitors in general have a short elimination-time. It therefore seems possible to continue treatment until shortly before surgery. The therapy should then be paused until proper wound-healing is achieved.

Ongoing development: Two selective JAK-1-Inhibitors are currently undergoing Phase 3 trials for clinical approval: Upadacitinib and Filgotinib. Both drugs show promising results in terms of efficacy and safety. Another JAK-inhibitor currently being evaluated in Phase 3 trials is the selective JAK-3-Inhibitor Peficitinib, with limited data available so far [80,81].

## 7. Conclusions

The therapeutic strategies of RA have improved dramatically over the past 3 decades. At that time, only a few drugs were available and therapy started late in disease course. With the T2T strategy and the possibility to choose from different mode of actions, the aim of stable remission can be reached and joint destruction can be prevented. Equally important to the choice of drugs is to diagnose the disease early and to start therapy within 6–12 weeks after disease onset. Early arthritis clinics headed by experienced rheumatologists are a good tool to achieve this. T2T further improves the prognosis of RA.

## Figures and Tables

**Figure 1 jcm-08-00938-f001:**
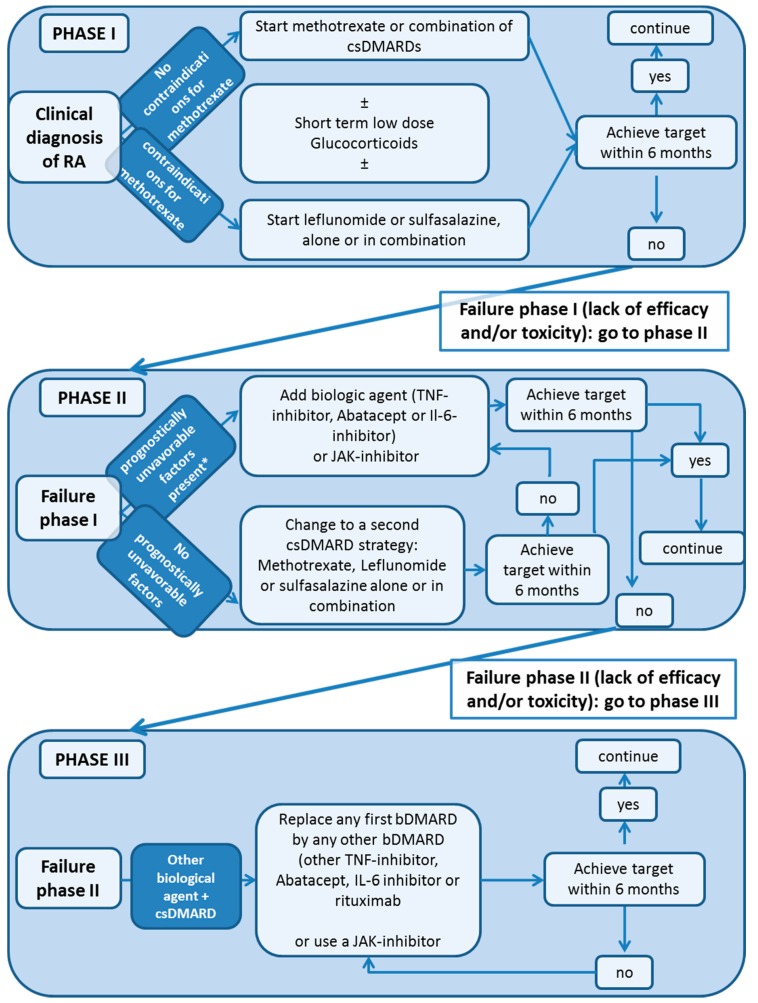
Algorithm adapted from the 2016 European League Against Rheumatism (EULAR) recommendationson rheumatoid arthritis (RA) management. bDMARD, biological; bsDMARD, biosimilar DMARDs; csDMARDs, conventional synthetic DMARDs; DMARDs disease modifying amtirheumatic drug; IL, Interleukin; MTX, methotrexate; TNF, tumor necrosis factor; tsDMARDs, targeted synthetic DMARDS.

**Figure 2 jcm-08-00938-f002:**
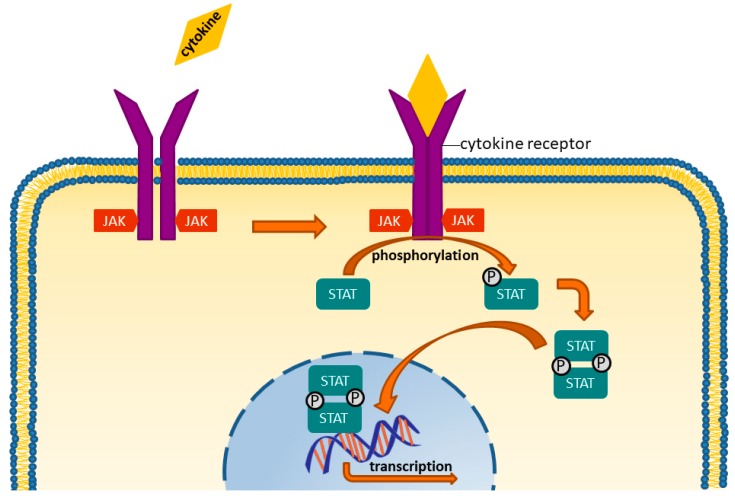
Cytokines bind to the receptors, which then undergo a change in their configuration into a dimeric structure, thereby activating their ability to work as a kinase. Consequently, they phosphorylate the STAT molecules (Signal Transducer and Activator of Transcription). Phosphorylated STATs also form dimeres that travel to the cell core, where they activate transcription processes, which further fuel inflammatory processes.

**Table 1 jcm-08-00938-t001:** Overarching principles of the T2T strategy (modified after Smolen J.S., et al. Ann Rheum Dis 2016;75:3–15). [3].

Overarching Principles of the T2T Strategy	
**1**	Basis for the treatment is a shared decision making between patient and doctor
**2**	Major treatment goals are: maximization of quality of life, normalisation of function and participation in social and professional life
**3**	The elimination of inflammation is essential to achieve the treatment goals
**4**	Outcomes in rheumatoid arthritis are improved by implementing T2T

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
