# Peer review of "Current Therapeutic Options in the Treatment of Rheumatoid Arthritis"

_jcm, 2019, doi:10.3390/jcm8070938_

Reviewer 1 Report

The article is interesting and sumarized the basics therapeutic options in the treatment of Rheumatoid Arthritis. However, I have some minor comments.

The title – correct the word „rheumatoide”

The Authors should add the Table which will be present all current therapise used in the treatment of RA patients, which is a conventional DMARD, which is a biologic, when was approved and where, adverse effect, mechanism of action ( drugs that reduce the effect of TNF-α, interleukin 6, a drug that inhibits the activation of T lymphocytes, that reduces the population of B cells, and that reduce the effect of interleukin 1).

Minor spell check is  required

Author Response

Dear Reviewer,

thanks for your advice.

The title – correct the word „rheumatoide”

I will correct it, thank you!

The Authors should add the Table which will be present all current therapise used in the treatment of RA patients, which is a conventional DMARD, which is a biologic, when was approved and where, adverse effect, mechanism of action ( drugs that reduce the effect of TNF-α, interleukin 6, a drug that inhibits the activation of T lymphocytes, that reduces the population of B cells, and that reduce the effect of interleukin 1).

Another table is a good idea! I will include if the Editor also think it is a good idea!

Minor spell check is  required

I will do a check before submitting the Remission.

Reviewer 2 Report

In their paper Koehler et al. reviewed the current strategies for the treatment of rheumatoid arthritis  with the special emphasis to T2T philosophy. Moreover they did the detailed description of currently available drugs and characterized their mode of action. The paper is clear and well written  and from the formal point of view only some minor comment should be addressed.

 In line 116 lymphocyte specific  should be used instead of German lyphozytespezific and again in line 369 tyrosine kinase should be written.

When describing currently available drugs in the light of side effect I would encourage the Authors to consider add some data on cardiological safety of TNF inhibitors which are contradicted in NYHA class III and IV stadium, however currently some data arisen suggesting that TNF blockade may bring different cardiovascular consequences in pure heart failure and rheumatoid arthritis.

Jak inhibitors is quite new therapeutic compound therefor the Authors should explain the clinical consequences  of various Jak blockade and how it translates to reduction of  executive cytokines.

The main concern is that is spite of the fat that is a well written review all data presented in the paper is well known that may significantly limit attention of the readers.

Author Response

Dear reviewer,

thank you for reviewing the paper.

In line 116 lymphocyte specific  should be used instead of German lyphozytespezific and again in line 369 tyrosine kinase should be written.

I corrected both.

When describing currently available drugs in the light of side effect I would encourage the Authors to consider add some data on cardiological safety of TNF inhibitors which are contradicted in NYHA class III and IV stadium, however currently some data arisen suggesting that TNF blockade may bring different cardiovascular consequences in pure heart failure and rheumatoid arthritis.

That is a good an important point. I will add another reference.

Jak inhibitors is quite new therapeutic compound therefor the Authors should explain the clinical consequences  of various Jak blockade and how it translates to reduction of  executive cytokines.

We tried to do this with our figure but maybe the figure Needs more explanation. JAK Inhibitors are new and therefore we tried to explain the mode of action in more detail in our revision.

Reviewer 3 Report

This manuscript is a brief description of the pharmacological treatment of rheumatoid arthritis without mentioning the basic principles of treatment, such as the withdrawal of tobacco, the control of obesity, etc ... nor treatment strategies that is a very important aspect and that must be mentioned, at least in the introduction

In page 3. The initial treatment with methotrexate should be in combination with glucocorticoids according to the update of the EULAR guidelines. Please comment on the results of the CARERA, t-REACH, CAMERA II or IMPROVE studies

In page 4 after first paragraph please comment risk factor for MTX failure in RA MTX-naïve patients

In page 5 first paragraph. MTX withdrawn before conception is a matter of discussion, please comment the effect in males (probably not necessary to withdrawn) and in women, as different recommendations depending on countries have been proposed

In page 6 last paragraph. Please add vaccinations according to recommendation before starting bDMARDs

Page7, line 200: please provide Ifx dose for RA

Page 7. Please comment immunogenicity as a side effect of biologics with important clinical consequences.

Page 7, line 223: Acute reactions may in some cases represent immunoglobulin E (please provide reference). In most cases acute reactions are due to immunocomplexes due to immunogenicity. B

In page 9 regarding IL-6 inh. Please add the most frequent side effects as infection, neutropenia, liver enzymes, increase in lipids, …

In page 11 for Rituximab please add that there is no indication for biologic naïve RA

In page 3, line 414 Tofacitinib being a JAK 1 and 3 inhibitor no anaemia is expected

Author Response

Dear reviewer,

Thank you for your suggestions.

This manuscript is a brief description of the pharmacological treatment of rheumatoid arthritis without mentioning the basic principles of treatment, such as the withdrawal of tobacco, the control of obesity, etc ... nor treatment strategies that is a very important aspect and that must be mentioned, at least in the introduction

You are absolute right. Nonpharmacological activities should be mentioned. I have done this in the Introduction now.

In page 3. The initial treatment with methotrexate should be in combination with glucocorticoids according to the update of the EULAR guidelines. Please comment on the results of the CARERA, t-REACH, CAMERA II or IMPROVE studies

I added this recommendation (page 4).

In page 4 after first paragraph please comment risk factor for MTX failure in RA MTX-naïve patients

I also added the risk factors (page 4)

In page 5 first paragraph. MTX withdrawn before conception is a matter of discussion, please comment the effect in males (probably not necessary to withdrawn) and in women, as different recommendations depending on countries have been proposed

I reword this paragraph and I hope I could clarify everything.

In page 6 last paragraph. Please add vaccinations according to recommendation before starting bDMARDs

I added this recommendation.

age7, line 200: please provide Ifx dose for RA

Page 7. Please comment immunogenicity as a side effect of biologics with important clinical consequences.

I added some information.

Page 7, line 223: Acute reactions may in some cases represent immunoglobulin E (please provide reference). In most cases acute reactions are due to immunocomplexes due to immunogenicity. B

I reword this text and added some other facts.

In page 9 regarding IL-6 inh. Please add the most frequent side effects as infection, neutropenia, liver enzymes, increase in lipids,

I did so.

In page 11 for Rituximab please add that there is no indication for biologic naïve RA

See line 338.

In page 3, line 414 Tofacitinib being a JAK 1 and 3 inhibitor no anaemia is expected

Anemia is deleted.

Thanks for your time and I hope you are satisfied with the change I made.

With kind regards

Round  2

Reviewer 2 Report

The paper looks fine I have no more comments .

Reviewer 3 Report

No more questions